# Extraordinary siblings: Mole rats, marmosets, and Radcliffe-Brown

**Doug Jones** *

Department of Anthropology, University of Utah, Salt Lake City, Utah, United States of America

* douglas.jones@anthro.utah.edu

## Abstract

According to the theory of kin selection, an organism that shows some level of altruism toward her kin – lowering her own fitness, raising that of a close genetic relative – may enjoy an evolutionary advantage. Some species show beyond-ordinary altruism toward siblings, and other kin, owing to unusual reproductive biology and/or ecology. Human beings are exceptional in another way: how we treat our kin depends partly on how we feel about them, but also partly on socially enforced norms. This article explores several versions of a simple evolutionary game, the Brothers Karamazov Game, that departs from the standard theory of kin selection to allow for the distinctively human capacity for establishing and enforcing social norms. We discuss possible applications to understanding the "unity of the sibling group" (Radcliffe-Brown) – according exceptional treatment to siblings, and to relatives classified as siblings or linked through siblings. We give special attention to lowland South America, where the sibling relationship is central to social organization.

## Introduction

Bonds between siblings are a common feature of both natural and human worlds. Evolutionary biologists have investigated the sibling tie under the heading of collateral altruism and the theory of kin selection. Social anthropologists increasingly recognize the sibling tie, along with parent-child and husband-wife ties, as a basic social building block in many small-scale societies.

This article considers the evolutionist's and the anthropologist's approaches to this topic, and explores their possible common ground. The article surveys exceptional versions of the sibling bond, among non-human animals ranging from social arthropods to mole rats and marmosets, and in human societies especially devoted to symbolically advertising and socially enforcing the solidarity and unity of the sibling group.

And, centrally, the article revises the theory of kin selection to allow for some of our species' exceptional qualities, especially our aptitude for attaining high levels of cooperation by devising, enforcing, and perpetuating social norms. This aptitude is important (so the argument goes) because it sometimes enables human beings to overcome the social dilemmas that occur when the cost of an action (e.g., fitting one's car with an anti-smog device) is borne by the actor, but the benefit (cleaner air) is more widely distributed. In cases like these, each member of a group might be better off if all paid the cost and all enjoyed the resulting collective benefit. The dilemma arises because each individual may be tempted to *free ride*, to avoid

**Data availability statement:** All relevant data are within the manuscript and its Supporting Information files.

1 / 20

**Funding:** The author(s) received no specific funding for this work.;

**Competing interests:** The authors have declared that no competing interests exist.

paying the individual cost while enjoying the benefits resulting from others' contributions. The dilemma may be resolved if some enforcement mechanism incentivizes individuals to contribute to the collective good [1].

Kin altruism poses a free rider problem in evolutionary form. An individual may get more of her genes into the next generation when she pays a fitness cost to provide a fitness benefit to a relative who shares her genes. This much is a familiar result from the theory of kin selection. But suppose there is a third party related to the other two. The altruist is creating a *positive externality* for this third party, meaning that the third party gains a benefit – more of copies of her genes pass into the next generation – without paying a direct cost. The third party would seem to be a free rider. And this holds reciprocally – the first altruist seemingly gets a free ride – whenever the third party pays a fitness cost to help their mutual kin. This suggests that two potential altruists could be better off, evolutionarily, if they made a bargain to work together rather than separately to help mutual kin in need.

To see how well this verbal argument holds up, this article explores some formal models of *socially enforced nepotism*, in which the *effective* levels of relatedness governing altruism toward kin are greater than in standard evolutionary models of individual altruism. The article improves on earlier work, deriving and comparing several formulas for *effective coefficients of relatedness* for collaborating altruists. The article argues that these simple models may offer insights into exceptional sibling ties in some human societies. What some other species attain through exceptional reproductive biology, human beings may attain through exceptional social abilities.

## Extraordinary siblings in evolutionary biology

To a first approximation, natural selection results in animals being adapted to leave as many surviving offspring as possible. This principle – each organism is built to maximize its fitness in its ancestral environments – provides an immediate explanation for why some social relationships have evolved. Consider parent-offspring ties: parents care for their offspring, at the expense of having fewer total offspring, if parental care results in more of their offspring surviving to reproduce. Or consider female-male relations: animals often display traits that impair their survival, when those traits are attractive to the opposite sex and contribute to the bearer's net reproduction. But adaptations may also spread in a population because they lead to increased offspring numbers, *not* among the carriers of those traits, but among relatives of the carriers. In its modern guise, the theory of kin selection enlists population genetics to give a numerical expression of this principle.

The theory can be formulated in several ways. One formulation treats kin selection as the outcome of selection at multiple levels [2,3]. At the higher level, selection operates among families (or other kin groups): families vary in their composition; because family members resemble one another there is more variation *between* families than there would be between groups assembled at random; those families with more members expressing pro-social, nepotistic traits produce more surviving offspring. At a lower level, selection also operates within families: within each family, those members expressing selfish, anti-social traits leave more offspring than other members. This *multilevel selection* among and within families allows for a moderate amount of pro-social behavior, short of perfect altruism.

Another formulation constructs a different two-way partition of the components of selection: each member of a family has, first, an individual, direct fitness component – the contribution she makes to her own survival and reproduction – and, second, an indirect fitness component – the contribution she makes to the survival and reproduction of family members who share her genes by recent descent [4]. Either of these channels of reproduction allows the organism to get her genes into the next generation. But since an organism shares only a

fraction of her genes with a relative (over and above what she shares with the population as a whole by remote descent), the indirect route is less effective.

Suppose there are tradeoffs between direct and indirect fitness. An altruistic organism may reduce her direct fitness in order to raise the fitness of one or more relatives. In that case, a gene or genes promoting altruism will be favored by kin selection if

$$\frac{c}{b} < r_H \qquad (1)$$

where $c$ is the cost of altruism to the altruist, $b$ is the benefit that the altruist provides to the beneficiary (possibly summed over multiple beneficiaries), and $r_H$ is the coefficient of relatedness, the expected number of genes in the beneficiary (or beneficiaries), given the number in the altruist, as a result of shared recent descent. More exactly, $r_H$ is a regression coefficient and $c$ and $b$ are measured in the currency of reproductive output.

For full siblings $r_H$ is equal to .5. If one sibling has inherited a gene for kin altruism from either parent, the probability that another sibling has also inherited the gene from that parent is .5 (assuming her parents are not related). A gene that leads one individual to lower her fitness by an amount $c$ will spread if it results in her raising the fitness of a sibling, $b$, by more than twice as much.

This is the usual case for siblings. Table 1 gives information about this case, and also about some groups of organisms that are of interest because they may have unusual coefficients of relatedness ($r$'s) among siblings, which may contribute to exceptionally strong sibling ties and functionally integrated kin groups. The non-human cases in the table provide a comparative benchmark for the human cases. For humans, one entry, in black, covers a case in which an unusual $r$ results from inbreeding, specifically first cousin marriage. Other entries for humans (cells shaded in blue) show something different. The $r$'s in these cases are *effective* coefficients of relatedness. These are greater than the standard $r$'s given by Hamilton's rule (unshaded

**Table 1. Coefficients of relatedness, Hamiltonian and effective.**

| Organisms | Rules of inheritance | Some (effective) $r$'s | Some social/ ecological factors |
|---|---|---|---|
| Many animals | Mendelian (diploid-diploid), no inbreeding | .5, full siblings, parent-child<br>.25, half siblings, g'parent-g'child<br>.125, cousins | family,<br>lifelong monogamy,<br>parental manipulation |
| Ants, Bees<br>various *Hymenoptera* | diploid female,<br>haploid male | .75, sister-sister<br>.5, mother-daughter<br>.25, sister-brother | shared nest,<br>reproductive suppression,<br>policing |
| Naked mole rats<br>*Heterocephalus glaber*<br>Social spiders,<br>various *Araneae* | inbreeding?<br><br>inbreeding | .81?, mean within colony<br><br>>.70 | shared nest, reproductive suppression,<br>policing,<br>communal webs |
| Marmosets<br>*Callithrix kuhlii* | chimerism | .574-.625,<br>twins | twinning,<br>paternal care |
| Humans<br>*Homo sapiens* | Mendelian,<br>no inbreeding | .67, 3 siblings<br>.75, 4 siblings<br>.22, 3 cousins<br>~1, $n \cdot r_H \gg 1$ | reputation + norms →<br>socially enforced nepotism |
| | inbreeding | .56, 2 siblings,<br>parents = cousins | reputation + norms →<br>marriage rules |
| | inbreeding,<br>intensive kinship | .72, 3 siblings,<br>parents = cousins<br>.17–.87, median .54<br>Amazonian horticulturalists, 24 societies | reputation + norms →<br>socially enforced nepotism +<br>marriage rules |

cells), implying that socially enforced nepotism in excess of individual altruism could be proportionately adaptive, as set out in the Methods section below.

Starting with the standard case in the first row, many animals form family groups with elevated levels of altruism among kin. Of particular note, an organism is equally related to a younger full sibling ($r = .5$) and to one of her offspring ($r = .5$). And a mother has an inclusive fitness interest in inducing her adult daughter to care for the daughter's juvenile siblings ($r = .5$ to mother) rather than the daughter's offspring ($r = .25$ to grandmother), e.g., by suppressing her daughter's reproduction [5]. The logic applies to either sex. This means that – given an initial condition of lifetime monogamous mating and overlapping generations – ecological conditions, possibly in conjunction with parental manipulation, can fairly readily result in some adult offspring staying as non-reproductive helpers at the nest rather than founding a new nest [6].

In some groups, exceptional coefficients of relatedness, with siblings related more closely than $r = .5$, may contribute to exceptional sibling altruism. Among ants and some bees (the next row in the table) multiple generations live together, with some individuals in each generation foregoing reproduction to rear siblings, a (maybe obligately) sterile all-female worker caste foraging and bringing food back to a nest. Ants and bees belong to the insect order *Hymenoptera*, and hymenopterans have an unusual *haplodiploid* system of reproduction: female offspring develop in standard fashion from the fusion of sperm and egg and have paired homologous chromosomes (*diploidy*), but males develop from an unfertilized egg and have only a single complement of chromosomes (*haploidy*). One result is that hymenopteran females are more closely related to their sisters ($r = .75$) than to their own offspring ($r = .5$). This should favor a hymenopteran female raising her mother's offspring rather than her own [4,7]. On the other hand, the high relatedness of a hymenopteran female to her sisters is countered by her low relatedness to her brothers ($r = .25$). In the simplest case, these factors cancel out, and there is no net advantage to a haplodiploid female in raising siblings rather than own offspring [8]. A more favorable result for exceptional sibling altruism may hold for a haplodiploid species with a *split sex ratio*, in which different categories of parents contribute systematically different sex ratios to the same generation [9,10]. Split sex ratios are not improbable; they may occur, for example, in the context of seasonality, reproduction by virgin queens, and/or queen replacement [10–12].

The literature on the topic is large and sometimes contentious [13–15], but many scholars believe that exceptional sibling relatedness has contributed to the evolution of exceptional sibling altruism among the *Hymenoptera*. One review concludes that "different modelling approaches concur that eusociality should evolve more easily in male haploid species than in those with both sexes diploid, for the same benefit and cost values" and "[t]he distribution of eusociality in insects is also consistent with male haploidy being a predisposing factor"[16]. Another review also acknowledges the possible contribution of haplodiploidy, while noting that "from a relatedness perspective, monogamy is likely to have been a more important driver of eusociality than the haplodiploidy effect" [11].

The next two rows in Table 1 list several groups of organisms (naked mole rats, social spiders) with high levels of sibling cooperation which are also reported as having unusual coefficients of relatedness, as a result of inbreeding. Naked mole rats live in colonies consisting of a breeding pair and many facultatively sterile workers of both sexes. They don't have to risk leaving their nest to find food, but build a network of tunnels that gives them access to underground tubers [17]. Along with special ecological conditions, unusual coefficients of relatedness may play a role in mole rat eusociality. Early work reported high levels of inbreeding resulting in within-colony coefficients of relatedness up to .81, although this may not be typical of the species [18,19].

Sociality in spiders [20,21] has evolved independently in 30 species (out of 47,000 described spider species), mostly in the tropics. Social spiders go beyond tolerating group members to cooperate in constructing large three-dimensional webs and in prey capture, colony defense, and alloparental care. They may have a division of labor, but without sterile castes. Social spiders mostly spend their adult lives in their natal webs, and the resultant high coefficients of inbreeding (mostly $r > .70$), are probably important in the evolution of high levels of cooperation (as well as female-biased sex ratios). But inbreeding also has long-term disadvantages: sociality in spiders seems to be an evolutionary dead-end.

One more row displays another case of exceptional coefficients of relatedness among siblings [22]. Marmosets develop as fraternal twins. Each twin is a *chimera*: while sharing the womb, twin embryos mingle cells, so that each incorporates cells, and genes, from the other. Chimerism extends to gonads and the germ line. When reproducing, a marmoset may be passing on a twin's genes, so any individual may be less related to her or his own offspring, and more closely related to her or his twin's offspring, than in the standard Mendelian case. Thus the effective $r$ among twin siblings is higher than the standard .5. It remains to be seen whether this leads to exceptional twin altruism.

In conclusion, theory implies that exceptional $r$'s can lead to exceptional sibling altruism (with ecological conditions and other features of biology also important), and evidence from various groups of non-human organisms supports this (with qualifications as noted above). The final rows in Table 1 are allotted to humans. Human beings are much like other animals in some ways, but quite extraordinary in others. The possible implications for kinship and sibling ties are the topic of the rest of this article.

## Extraordinary human siblings

What is exceptional about the sibling tie in humans?

The basic biology of siblinghood in our species is not unusual. Inheritance in human beings follows standard Mendelian rules: no haplodiploidy, no chimerism. The level of inbreeding in human populations mostly varies from low to moderate (but see below on endogamy and inbreeding).

Nor does the sibling tie in our species entail exceptional individual altruism. Human societies do not approach the level of reproductive division of labor found in some eusocial insects, where members of obligately sterile specialized castes labor to provision and defend a much smaller number of their reproductive siblings in vast colonies that amount to functionally integrated superorganisms.

However the human sibling tie is exceptional in other ways. Sibling relations vary strikingly from one society to another. And this variation is enabled by an exceptional human quality: the social regulation of kinship according to culturally transmitted norms. Below, we consider the centrality of social norms to human behavior in general, and then consider several ways in which socially enforced rules can affect the sibling tie.

Next, in the Methods section (and in two supplements) we work through a simple model of sibling altruism involving three (or more) siblings. We demonstrate that when altruism toward siblings is enforced by third (and fourth, etc.) parties, the effective coefficient of relatedness governing kin altruism can be greater than the standard Hamiltonian $r$. We also consider more distant relations. Kin groups can bring these relations closer, enfolding distant kin in a sibling ethic. They can achieve this by using social pressure to raise effective coefficients of relatedness to more distant kin, and by manipulating marriage rules to raise Hamiltonian $r$'s.

We begin, then, with some general considerations about human sociality. Evolutionary biologists have their theories of sociality, which include a theory of kin altruism; the theory applies very broadly, provided we are careful with our $r$'s, $b$'s and $c$'s. But social anthropologists

have argued from early on that in theorizing about human sociality, we must reckon with something distinctively human, an outsized aptitude for making, enforcing, and perpetuating social norms.

Durkheim coined the expression *homo duplex* for the resulting dualism of structure and sentiment [23]. And here is Radcliffe-Brown [24]:

> Every human being living in society is two things: he is an individual and also a person. As an individual, he is a biological organism … The human being as a person is a complex of social relationships.

The same idea is expressed by the *dramaturgic metaphor* [25,26]: all the social world's a stage, and how the men and women on't treat one another depends not only on how they feel about one another, but also on the social roles they play, and the attendant social scripts. Some roles and some occasions are tightly scripted, others allow more room for improvisation and individual personality. Some societies are relatively *tight* – they demand that rules be followed closely – others are relatively *loose* [27]. Within societies, some relationships, especially those with superiors or more distant others may be particularly formal and respectful. Expected proper behavior is often tied to kinship roles. Across cultures, famously, relations between a man and his mother-in-law are often hedged about with respect and taboo. Other kinship roles may veer in the opposite direction, with joking and horseplay allowed and expected [28,29].

The social regulation of behavior operates through reputation. Reputation takes on particular importance in human societies, thanks in large part to language [30], and to communication and cognition about affairs beyond the immediate here-and-now. In face-to-face communities, each person has not only a direct reputation, based on what others witness, but an indirect reputation, based on what others say about her [31]. Conversation between A and B about C is an important avenue for policing behavior. Reputation and reputation management have likely had fitness consequences and long-term evolutionary effects, resulting in an animal zealous in enforcing norms and fearful of flouting them [32]. Evolutionary theory here converges with sociological tradition.

"We are 90 percent chimp and 10 percent bee" [33]. Human beings rarely show anything like the unforced extreme altruism of ants and honeybees. But the normative regulation of social behavior allows people to mimic social insects in one respect: human beings can build functionally integrated societies on a scale of thousands or millions of actors [34]. And the same "glue" – interlocking identities, roles, norms, and scripts – that holds together large-scale impersonal organizations is also at work on smaller scales [35]. This may have implications for human kinship systems and sibling ties.

## Methods

Consider how we might modify the theory of kin selection to allow for more in the way of social rule-following. The standard exposition of the theory of kin selection assumes the barest minimum of social structure: it starts with an actor who, on her own, acts or doesn't act to help a sibling or other kin; the beneficiaries are passive recipients of assistance. Here we are interested in a more complicated setup, in which strategic interaction with third parties affects an individual's behavior toward needy kin. This turns a one-actor optimization problem ("Maximize your inclusive fitness!") into a problem in multi-actor game theory.

Following common practice in evolutionary modelling, we resort to a drastic simplification of reality to capture some of the underlying forces at work, considering the evolutionary dynamics of altruism with three full siblings, two of whom are involved in helping a third.

We derive and compare several versions of the effective coefficient of relatedness (effective $r$). While the standard Hamiltonian $r = .5$ dictates the level of altruism (maximum cost-benefit ratio for each player) favored by natural selection when helpers act independently to benefit a needy sibling (Condition 1), the effective $r$'s below operate when two siblings act together.

The discussion is organized as follows

**The Brothers Karamazov Game.** This game was introduced in a previous publication [36] (See also [37].). Here we amend the game by allowing for individual differences in helping ability. This elaboration allows for different versions of effective $r$ and for evolutionary dynamics with a range of possible strategies.

**Version 1.** We follow standard population genetics to derive an effective $r$ (Condition 2.0) greater than the standard $r$.

*Excursus, concerning* p. The effective $r$ above has a frequency dependent term, $p$, which is absent from the standard $r$, implying that socially enforced nepotism works best when it's at high frequency. We show where this frequency dependence comes from (Condition 2.1).

**Version 2.** When we introduce other, more flexible strategies that are sensitive to the helping abilities of other players, the resulting effective $r$ depends on players' abilities, but not on the frequencies of different strategies (Condition 3).

**Diploidy.** For ease of exposition, the games above assume haploid players. A diploid version of the game also gives Condition 3. The derivation is given in an online supplement.

*Competing strategies*. When the inflexible players of Condition 2 compete with the flexible players of Condition 3, the latter win, making the frequency-independent Condition 3 a preferred form of the effective $r$.

**Version 3.** When the game is played for multiple rounds, with the option of "cheating" on any round, the effective $r$ further depends on the expected number of rounds played (Condition 4). The derivation is given in an online supplement.

**In summary.** For readers who want to skip the derivations, the most important results – the winning effective $r$'s given by Conditions 3 and 4 – are reviewed here.

**The Brothers Karamazov Game.** Imagine a group of three full siblings, Ivan, Alyosha, and Dmitri. Suppose that Dmitri often finds himself in trouble, and turns to one of his brothers for help, turning half the time to Ivan and half to Alyosha. Dmitri is never in a position to reciprocate. We are interested in the fate of a gene for altruism, a gene that leads either Ivan or Alyosha to benefit Dmitri at some cost to themselves, where benefits and costs are measured in the currency of biological fitness.

If Ivan and Alyosha act independently of one another, then the mathematics involved is that of the standard Hamilton's rule. A gene for altruism will be favored if Ivan helps Dmitri whenever his cost is less than half the benefit to Dmitri (because there is an expected .5 chance that Dmitri has also inherited the gene from one of their parents). And similarly for Alyosha helping Dmitri.

But suppose we introduce more in the way of social interaction to the model. Suppose that Ivan makes a conditional offer to Alyosha. Whatever Alyosha does to help Dmitri, Ivan will match it. His offer allows for differences in their abilities, which are known to both. When Ivan and Alyosha have abilities $\alpha_I$ and $\alpha_A$ respectively, Ivan offers to make his benefits and costs equal to $\alpha_I \cdot b$ and $\alpha_I \cdot c$ if Alyosha makes his benefits and costs equal to $\alpha_A \cdot b$ and $\alpha_I \cdot c$ for some $b$ and $c$.

We consider several versions of the game, each giving an effective coefficient of relatedness larger than the standard Hamiltonian .5 for siblings. Readers may wish to skip ahead to Version 2, which supersedes Version 1, or skip all the way to Conditions 3 and 4.

**Version 1.** This version is laid out in Table 2, which gives all combinations of two alleles, G, for conditional nepotism, and H, for no conditional nepotism, for haploid full siblings

**Table 2. Brothers Karamazov Game, Version 1.**

| Ivan $\alpha_I$ | Alyosha $\alpha_A$ | Dmitri $\alpha_D = 0$ | frequency | ΔG | ΔH |
|---|---|---|---|---|---|
| G | G | G | $p^2 + \dfrac{1}{4}pq$ | $(\alpha_I+\alpha_A)\cdot(b-c)$ | 0 |
| G | G | H | $\dfrac{1}{4}pq$ | $-(\alpha_I+\alpha_A)\cdot c$ | $(\alpha_I+\alpha_A)\cdot b$ |
| G | H | G | $\dfrac{1}{4}pq$ | 0 | 0 |
| G | H | H | $\dfrac{1}{4}pq$ | 0 | 0 |
| H | G | G | $\dfrac{1}{4}pq$ | 0 | 0 |
| H | G | H | $\dfrac{1}{4}pq$ | 0 | 0 |
| H | H | G | $\dfrac{1}{4}pq$ | 0 | 0 |
| H | H | H | $q^2 + \dfrac{1}{4}pq$ | 0 | 0 |

Ivan, Alyosha, and Dmitri. This implies a life cycle in which haploid gametes unite in a brief diploid phase, immediately go through meiosis, and give rise to a haploid multicellular stage. In humans of course the multicellular part of the life cycle is diploid; we consider a diploid case in a supplement.

The brothers' abilities are respectively $\alpha_I$, $\alpha_A$, and 0. The table shows the expected frequencies of each combination, where G and H have population frequencies of $p$ and $q$, with $p + q = 1$. Mating is random. There is a probability $p^2$ that both parents are G, so the three brothers are all G. There is probability $q^2$ that both parents are H, so the three brothers are all H. And there is probability $2pq$ that one parent is G and the other H, so each of the eight offspring genotype combinations is equally likely.

The table also shows benefits and costs (multiples of $b$ and $c$). Extra nepotism happens if and only if both Ivan and Alyosha have genotype G. In this case, benefits given by Ivan and Alyosha to Dmitri are $\alpha_I \cdot b$ and $\alpha_A \cdot b$ respectively, and costs $\alpha_I \cdot c$ and $\alpha_A \cdot c$, for some $b$ and $c$. In other words, the cost/benefit ratio is the same for the two brothers. (In an alternative formulation, costs and benefits vary continuously, and Ivan and Alyosha equalize marginal returns, $db/dc$, rather than cost/benefit ratios, according to their abilities. Results in that case are comparable to those given here.)

When Ivan, Alyosha, and Dmitri all have genotypes G, Ivan pays cost $\alpha_I \cdot c$ to give Dmitri benefit $\alpha_I \cdot b$ and Alyosha pays cost $\alpha_A \cdot c$ to give Dmitri benefit $\alpha_A \cdot b$, and the absolute increment in fitness for allele G, ΔG, is given by $(\alpha_I+\alpha_A)\cdot b - (\alpha_I+\alpha_A)\cdot c$. When Ivan and Alyosha have genotype G and Dmitri has genotype H, the result is a cost $(\alpha_I+\alpha_A)\cdot c$ for allele G and a benefit $(\alpha_I+\alpha_A)\cdot b$ for allele H. In all other cases, Ivan and Alyosha fail to reach an agreement to pay extra costs. While individuals with genotype H may still carry out individual unconditional nepotism, we are interested in costs and benefits over and above the standard Hamiltonian $c/b < .5$, so costs and benefits in the unconditional cases are normalized to 0.

We can multiply the frequency of each genotype combination in Table 2 by the associated ΔG in the same row, and then add these products, to get the absolute increase in fitness of G

summed over all combinations, and similarly for H. To get the rate of increase (normalized fitness) for G and H, we divide these sums by the frequency of each type, p and q. Solving for $c/b$, the rate of increase for G is greater than the rate for H when $c/b < r_G$ where:

$$r_G = \frac{1+2p}{2+2p} \tag{2.0}$$

When $p = 1$, the effective coefficient of relatedness $r_G$ is equal to .75. An established conditional nepotist strategy with $r_G = .75$ can resist invasion by a Hamiltonian nepotist with $r_H = .5$ or a conditional nepotist with effective coefficient of relatedness less than .75.

*Excursus, concerning* p. The conditional coefficient of relatedness, $r_G$, includes a gene frequency term, $p$, which is not part of the usual theory of kin selection. We can see where this frequency dependence comes from, and get an inclusive-fitness-centered gene's-eye view of $r_G$ as follows:

Take an individual donor, Ivan or Alyosha. He is agreeing to a package deal in which Dmitri receives a benefit of $(\alpha_I + \alpha_A) \cdot b$. Since Dmitri is a brother to the selected donor, we discount the gains to Dmitri by the standard coefficient of relatedness, .5, so the value of the package to the donor is $.5(\alpha_I + \alpha_A) \cdot b$. The donor is paying a cost, either $\alpha_I \cdot c$ or $\alpha_A \cdot c$.

This donor also has to take into account the cost paid by the other donor, discounted by his relatedness to the other donor. Here the calculation gets tricky. We can't use the familiar pedigree coefficient of relatedness among brothers. Instead we have to calculate the coefficient of relatedness – the probability of having genes identical by descent – *conditional on the two potential donors agreeing to the deal.*

Table 3 shows how this works.

We assume that one of the brothers' parents has a copy of the gene for conditional altruism. We mark this copy with an underline, G. The genotype of the other parent is unknown, shown as U. The state of U is G with probability $p$ or H with probability $q$.

If Ivan and Alyosha both inherit G, they are identical by descent (IBD) at this locus, and with probability 1 they make a deal to help Dmitri. If one inherits G and the other inherits U, they are not IBD, and they help Dmitri with probability $p$.

We can rewrite Condition 2.0 as:

$$\frac{1}{r_G} = 1 + \frac{1}{1+2p} \tag{2.1}$$

The two terms in the sum on the righthand side of the equation correspond to two components in the inclusive fitness calculation of our donor. The 1 term corresponds to the cost paid by the donor. The $1/(1+2p)$ term corresponds to the cost paid by the other donor, discounted by the relatedness to the other donor, the probability of IBD for G conditional on the two potential donors agreeing to the deal.

**Table 3. Concerning p.**

| Ivan | Alyosha | Ivan & Alyosha IBD | Prob[Help Dmitri] |
|------|---------|--------------------|--------------------|
| G | G | 1 | 1 |
| G | U | 0 | p |
| U | G | 0 | p |
| U | U | n.a. | n.a. |

If G is rare, $p \approx 0$, the only way Ivan and Alyosha are likely to share G is through identity by descent from one parent. Then $1/(1+2p) \approx 1$ and $r_G \approx .5$. Cost to a brother counts the same as cost to oneself, and the advantage to pooling nepotism disappears.

But if G is common, $p \approx 1$, Ivan and Alyosha will probably agree to help Dmitri; they are likely to share G, even if they are not IBD for G. Then $1/(1+2p) \approx .33$ and $r_G \approx .75$. The inclusive fitness cost of conditional nepotism is less, and the potential for conditional nepotism is greater.

In short, the Brothers Karamazov Game, Version 1, demonstrates that conditional nepotism can work. But it also provides a reminder that the coefficients of relatedness that matter for calculations of inclusive fitness may differ from the standard genealogical $r$'s. If Alyosha agrees to a deal with Ivan, this tells Ivan something about Alyosha and his genotype beyond the fact that Alyosha is a brother. Ivan can use this information and act accordingly.

**Version 2.** The result above assumes rigid behavior on the part of both types when abilities differ. Above, a more able type G always offers full conditional nepotism even when the less able party would bear little of the extra cost. And a less able type H always declines to contribute to extra nepotism even when the more able party would bear most of the cost.

Suppose we introduce more flexible behavior to the Brothers Karamazov Game. Imagine an alternative version of H, a player who plays by Hamilton's rule, $c/b < .5$, as long as the other party does so, but who is also willing to cooperate in providing extra benefits, as long as the other party is more able and bears most of the cost. (In the knife-edge case in which abilities are equal, imagine H cooperates with G with probability .5, but never cooperates with another H.) Imagine further that G's behavior adjusts accordingly.

Table 4 presents this alternative assumption. (The more able party is given as Ivan, but obviously we could switch and make Alyosha more able.)

Solving, as above, for $c/b < r_G$ in this case gives

$$r_G = \frac{\frac{1}{2}\left(\alpha_I + \alpha_A\right)}{\alpha_I + \frac{1}{2}\alpha_A} \tag{3}$$

Condition 3 implies that $r_G$ goes from .5 when $\alpha_A$ is equal to 0, to .67 when $\alpha_A$ is equal to $\alpha_I$. When $\alpha_A$ is equal to zero, Alyosha is unable to offer any help, and we recover the standard Hamilton's rule condition for independent altruism. But when $\alpha_A$ is greater than zero, both Ivan and Alyosha treat Dmitri as if he were even closer than a full sibling.

This version of $r_G$ has a straightforward inclusive fitness interpretation. Consider the accounting for Ivan, assumed to be the more able donor. He is offering a package deal to Alyosha. The benefits to be provided by the two are proportional to $\alpha_I + \alpha_A$. Discounted by the standard coefficient of relatedness for siblings, this gives the numerator, $\frac{1}{2}(\alpha_I + \alpha_A)$. The costs to be incurred are proportional to $\alpha_I$ for Ivan and $\alpha_A$ for Alyosha. Ivan can't know whether Alyosha is G or H, since either type will agree to the deal. Because Ivan has no information about Alyosha's genotype beyond the fact that he is a brother, he discounts Alyosha's costs by the standard .5 for a sibling. This gives the denominator $\alpha_I + \frac{1}{2}\alpha_A$.

Here's another way to think about the difference between this version and Version 1 of the Brothers Karamazov Game. In Version 2, expected benefits and costs are a linear function of the number of copies of G among the donors. With two copies of G between Ivan and Alyosha, full costs are paid and benefits gained. With one copy of G and one of H between the two, the probability is .5 that the more able of the two is G and the deal goes through and .5 that the more able is H and no deal is made. This is in contrast to Version 1 of the game, in which

**Table 4. Brothers Karamazov Game, Version 2.**

| Ivan $\alpha_I$ | Alyosha $\alpha_A < \alpha_I$ | Dmitri $\alpha_D = 0$ | frequency | $\Delta G$ | $\Delta H$ |
|---|---|---|---|---|---|
| G | G | G | $p^2 + \frac{1}{4}pq$ | $(\alpha_I + \alpha_A) \cdot (b-c)$ | 0 |
| G | G | H | $\frac{1}{4}pq$ | $-(\alpha_I + \alpha_A) \cdot c$ | $(\alpha_I + \alpha_A) \cdot b$ |
| G | H | G | $\frac{1}{4}pq$ | $(\alpha_I + \alpha_A) \cdot b - \alpha_I \cdot c$ | $-\alpha_A \cdot c$ |
| G | H | H | $\frac{1}{4}pq$ | $-\alpha_I \cdot c$ | $(\alpha_I + \alpha_A) \cdot b - \alpha_A \cdot c$ |
| H | G | G | $\frac{1}{4}pq$ | 0 | 0 |
| H | G | H | $\frac{1}{4}pq$ | 0 | 0 |
| H | H | G | $\frac{1}{4}pq$ | 0 | 0 |
| H | H | H | $q^2 + \frac{1}{4}pq$ | 0 | 0 |

benefits and costs are a non-linear function of the number of would-be conditional altruists, and the inclusive fitness calculation is more complicated.

**Diploidy.** Condition 3 also holds for a diploid version of Version 2, more appropriate for the humans, assuming heterozygotes play strategy H with probability .5 and G with probability .5. This is shown in an online supplement.

*Competing strategies.* Putting Version 1 and 2 of the Brothers Karamazov Game together implies the following evolutionary dynamics. Suppose that $H_1$ and $G_1$ play the game according to Version 1 rules, and $H_2$ and $G_2$ play according to Version 2.

If we start with just $H_1$ and $G_1$, then

- A population of $H_1$, effective $r = .5$, resists invasion by $G_1$, effective $r = .75$, but

- A population of $G_1$, effective $r = .75$, resists invasion by $H_1$, effective $r = .5$.

But if we introduce $H_2$ to a population of $G_1$, then

- $H_2$, effective $r = .5$, successfully invades a population of $G_1$, effective $r = .75$.

And if we then introduce $G_2$

- $G_2$, effective $r$ between .5 and .67 depending on $\alpha_I$ and $\alpha_A$, successfully invades a population of $H_2$, effective $r = .5$.

So $G_2$ is the winner. This does not depend on the frequency of the $G_2$ allele.

**Version 3.** This version is treated more fully in another online supplement. We summarize results here.

Above, we assume binding agreements between Ivan and Alyosha. Suppose we relax this assumption. and introduce a variant "cheater" strategy, $H_3$. When Alyosha, the weaker of the two active players, is $H_3$, he plays the same as $H_2$ above. However when Ivan, the stronger

of two active players, is $H_3$, he pledges to match Alyosha's contribution, but then breaks his promise, leaving Alyosha with cost $\alpha_A \cdot c$, Dmitri with benefit $\alpha_A \cdot b$, and Ivan with cost 0. This strategy, which amounts to Ivan playing All Defect in a version of a Prisoner's Dilemma Game, will beat the previous winner, $G_2$. (Alternative cheater strategies, with Alyosha duping Ivan, are less invasive than $H_3$.)

But now let's make the game a Repeated Prisoner's Dilemma Game, where Ivan and Alyosha may have further opportunities to help Dmitri. Suppose that, after any one round of helping, the probability that Ivan and Alyosha will play another round is $w$, so the expected number of rounds played is $1 + w + w^2 + w^3 + \ldots$ or $1/(1-w)$. And now introduce a variant $G_3$ who plays a Tit For Tat strategy, beginning by playing like $G_2$, then punishing any defection by the other player with a defection on the next round, but otherwise reciprocating in proportion to appropriate $\alpha$'s. A population of $G_3$ is resistant to invasion by $H_3$ as long as $c/b < r_G$ where

$$r_G = \frac{\frac{1}{2}\left(\alpha_I + w\alpha_A\right)}{\alpha_I + \frac{1}{2}w\alpha_A} \qquad (4)$$

For a single round of play, i.e., $w = 0$, $r_G$ is .5. As the probability of playing another round goes to 1 and the expected number of rounds played goes to infinity, we recover Condition [3] above.

**In summary.** The preferred, winning versions of the effective coefficient of relatedness are given by Condition [3] (when Ivan and Alyosha can make a deal and stick to it) and Condition [4] (when either player can "cheat" on a deal and the game is played for multiple rounds). Condition [3] implies that conditional altruism attains a maximum, with effective $r = .67$, when Ivan and Alyosha have equal helping abilities, meaning that each of the two is willing to pay a fitness cost of up to 2 to provide a fitness benefit of 3 to Dmitri, provided the other does the same. It reaches a minimum, $r = .5$, when the weaker partner has no ability to help. This is the standard result, in which a sibling acting alone is willing to pay a fitness cost of up to 1 to provide a fitness benefit of 2 to another sibling. Condition [4] implies that as the expected number of rounds played approaches infinity, effective $r$ approaches Condition [3]; with one round of play, cheating prevails and effective $r$ equals .5.

Both Condition [3] and Condition [4] of the Brothers Karamazov Game imply that getting started at a low frequency and increasing to high frequency – difficult in the case of reciprocity among non-relatives – is relatively easy in the case of socially enforced nepotism among siblings.

## Results

Ivan and Alyosha may treat Dmitri as even closer than a full sibling. This is possible because altruism toward kin is a *public good*. When one individual incurs a cost to help a relative, she is also providing a free inclusive fitness benefit to any other relatives of the beneficiary. This fits a standard criterion of a public good, the benefit is *non-excludable*; by the very nature of inheritance, a player can't avoid providing an inclusive fitness benefit to other relatives. Thus, as with other public goods, more of the good (benefits to a needy relative) is provided when benefactors act together rather than separately [38].

(In the setup we consider here, the benefit also fits the other criterion of a public good, it is *non-rivalrous*; one player providing a benefit doesn't prevent another player from also providing a benefit. The non-rivalrous/rivalrous distinction is less important here. We could cover the rivalrous case by having Ivan and Alyosha each pay a fraction of the cost of a rivalrous benefit rather than providing separate benefits, with similar final results.)

Evolutionary theory offers two different frameworks for interpreting what's going on with Conditions [3] and [4].

In the multilevel selection framework, Ivan and Alyosha together, especially when they engage repeatedly, are a collective actor. The level of altruism that they agree on is given by $r_G$. This is a group coefficient of relatedness, the relatedness of Ivan+Alyosha to Dmitri when they act together rather than separately.

In the inclusive fitness framework, Ivan incurs a cost $c_{ID}$; this cost directly benefits Dmitri by $b_{ID}$, and also "buys" a matching contribution from Alyosha which costs Alyosha $c_{AD}$ and benefits Dmitri $b_{AD}$. When Ivan counts up his inclusive fitness costs and benefits, he counts his full cost. He also counts half Alyosha's cost and half the summed benefits to Dmitri: just half since each of his siblings has just a .5 chance of inheriting the conditional altruism gene.

## Beyond three brothers

We could elaborate this simple model in various ways. For example, we could allow costs and benefits to vary continuously, with Ivan and Alyosha equating their marginal returns, *db/dc*, rather than the ratios *b/c*. Also a more general model of the Brothers Karamazov Game could assign a helping ability to Dmitri as well as to the other two, and would allow reciprocity between each pair, so that, for example, how much Ivan helps Alyosha could depend both on how much Alyosha helps Dmitri and on how much Alyosha helps Ivan, and so on. This would allow a range of possibilities, from joint altruism, as considered here, to pure reciprocity with equal abilities for the three brothers. And players might have either perfect or imperfect information about one another's abilities.

These are topics for the future. Here, we stick with a different elaboration, adding helpers. With four full brothers, where Ivan, Alyosha, and Smerdyakov act jointly to help Dmitri, we get:

$$r_C = \frac{\frac{1}{2}\left(\alpha_I + \alpha_A + \alpha_S\right)}{\alpha_I + \frac{1}{2}\left(\alpha_A + \alpha_S\right)} \tag{5}$$

where the helpers' respective abilities are $\alpha_I$, $\alpha_A$, and $\alpha_S$, and $\alpha_I \geq \alpha_A$, $\alpha_I \geq \alpha_S$. In this case, the effective coefficient of relatedness, $r_G$, reaches .75 when $\alpha_I = \alpha_A = \alpha_S$. More generally, as the number of helpful siblings increases, $r_G$ approaches 1.

We can also consider more distant relations. If we make Ivan, Alyosha, and Dmitri full first cousins, we get

$$r_C = \frac{\frac{1}{8}\left(\alpha_I + \alpha_A\right)}{\alpha_I + \frac{1}{8}\alpha_A} \tag{6}$$

with $r_C$, $\alpha_I$, $\alpha_A$ as before. This reaches .22 when $\alpha_I = \alpha_A$, which is almost twice the individual coefficient of relatedness for cousins, .125, and close to the individual coefficient of relatedness for half-siblings, .25.

Versions of the game can be extended even further, to larger and more distantly related kin groups [36,38]. If the number of helpers, *n*, is large relative to the reciprocal of the individual coefficient of relatedness, $1/r_H$, i.e.,

$$n \cdot r_H >> 1 \tag{7}$$

then the effective coefficient of relatedness approaches 1.

For large *n*, the exact value of the effective coefficient of relatedness, $r_G$, depends on how collective decisions are made and enforced. But the same logic applies: altruism toward kin is a public good. To the extent that members of a large group act collectively to help their kin, they can attain a high effective coefficient of relatedness even if their individual Hamiltonian *r*'s are small. Socially enforced norms can turbo-charge altruism toward kin.

These results are summarized in Table 1 (Humans/ Mendelian, no inbreeding), the first set of numbers in blue.

## Discussion

### Extraordinary siblings in social anthropology

In a classic statement on the anthropology of siblinghood, Radcliffe-Brown [24] writes

> The bond uniting brothers and sisters together into a social group is everywhere regarded as important, but it is more emphasized in some societies than in others. The solidarity of the sibling group is shown in the first instance in the social relations between its members.

I suggest that the Brothers Karamazov Game, simple as it is, offers an elementary model of the solidarity of the sibling group. It encompasses variability in the strength of the sibling tie, but does not depend on an exceptional mode of inheritance. Human beings can attain more-than-marmoset-level *effective* coefficients of relatedness even without marmosets' unusual biology.

I suggest further that the extension of the Brothers Karamazov Game to more, and more distant, kin provides a model for a further principle [24].

> From [the solidarity of the sibling group] there is derived a further principle... that of the unity of the sibling group... its unity in relation to a person outside it and connected with it by a special relation to one of its members.

An organizing principle of many societies is to shrink the social distance between genealogically remote kin by equating siblings, especially same-sex siblings, treating (at least ideally) one sibling in a sibling set as substitutable for, or equivalent to, another. This principle is expressed in a variety of social institutions and normative practices, including mutual aid to needy kin; sharing of, and collective claims on, food, territory, and other property; rules of inheritance including fraternal inheritance; joint defense, including obligations of vengeance; collective liability for, and claims to, marriage payments; and marriage rules, including the levirate, sororate, and prescriptive cousin marriage [24,39–41].

The unity of the sibling group is commonly reinforced by symbolic expressions, including rites of passage and other costly rituals, conception beliefs, and social memory in the form of genealogies and mythical charters. The unity of the sibling group is often expressed in kin terminology, as in bifurcate merging terminologies in which an individual's mother and mother's sister are called by the same kin term, and her father and her father's brother. When a sibling set is treated as a unit, relatives related through those siblings are brought closer. When this is followed consistently, cousins, especially parallel cousins, may be equated with siblings, and this extension may apply to even more remote relations.

The recurring theme is a determination to extend a sibling ethic to more distant relations. This is consistent with the argument here that social enforcement can bring the effective coefficient of relatedness for distant kin into the range of sibling relatedness. But the ethnographic evidence is also consistent with another part of the argument: the unity of the sibling group, and the associated axiom of amity, is a socially promoted ideal, which can fail in practice

The solidarity and unity of the sibling group, where they are important, play out differently in different societies. Below we consider how two further principles – descent and alliance, expanding on parent-child and husband-wife ties – relate to the present work. We give special attention to lowland South America. Anthropologists have noted that the sibling tie takes on special importance as a social building block in this culture area. One anthropologist notes [42]:

> [T]he significance of siblingship has emerged from consideration of specific topics and specific ethnographic cases – not from ethnographic first principles. Siblingship has not been emphasized because ethnographers sought to employ pre-existing theoretical constructs, but because it is so ethnographically prominent that it is difficult to miss or avoid … siblingship shows up everywhere but remains relatively underdeveloped theoretically.

This article has developed some novel theoretical constructs relating to siblingship: we conclude with a return to Table 1, to propose that exceptional effective *r*'s, resulting from a combination of inbreeding and socially enforced nepotism, may foster exceptional sibling ties in our species.

## Descent, alliance, siblinghood, and lowland South America

In classic descent theory [43] a descent group is a corporation holding an estate, which consists of rights in things and/or persons. Unilineal descent assigns each individual to one and only one descent group at a given genealogical depth. Descent grouping can function as an alternative to organization on the basis of territorial divisions, characteristic of many modern state societies.

For example, among Indonesian Lamaleran whale hunters, social kinship – membership in patrilineal clans – rather than individual genetic kinship is the basis for organizing whaling crews, although after the initial clan-based distribution of whale meat, biological kinship governs the further sharing between households [44,45]. If we allow that the overlapping kinship ties characteristic of Lamaleran descent groups can amplify effective coefficients of relatedness, this fits the argument here that kinship can involve both culturally particular, socially enforced nepotism, and universal individual attachments to close kin.

A different evolution-minded theory proposes instead that a descent group is the expression of its founder's *descendant-leaving strategy* [46–48]. It is in the fitness interests of powerful individuals to induce their descendants to cooperate with one another more than they otherwise would. The founders of lineages and clans may engineer institutions and values that perpetuate such cooperation over generations – an enduring *post mortem* elaboration of altruism-as-parental-manipulation. Perhaps consistent with this hypothesis is population genetic evidence that privileged patrilineages have experienced long term reproductive advantages in post-Neolithic Eurasia [49].

However, the theory that descent groups express the extended phenotypes of Great Men (and occasional Great Women) raises some questions. What mechanisms sustain cooperation among the descendants of a posthumous founder? The hypothesis here is that the persistence of elevated nepotism depends on social enforcement by the living, acting on their collective interest in solving a public goods problem inherent in kin altruism.

A further issue is that descent theory, whether in its classic or evolutionary formulation, seems to be a poor fit for how kin groups actually work in many societies [50–53]. For example, in lowland South America, "[d]istinctions that play… a central role in the social anthropological literature on descent – notably, emphasis on the corporate properties of descent

groups, on genealogical reckoning, and on the analytic distinction between descent and patrifiliation – seem… of dubious relevance" [54].

The limitations of descent theory have spurred the search for alternatives. One of these is alliance theory. In the classic formulation [55] kinship is structured around marriage rules. These, directly or circuitously, are systems of reciprocity between wife's kin and husband's kin. In a simple direct marriage exchange, two brothers marry one another's sisters. When the reciprocal relations thus established are continued in the next generation, the result is two bilateral cross-cousin marriages. An evolutionist take on this scheme adds nuclear family inbreeding avoidance as a motive for exogamy [56].

As with descent theory, this model fits some societies better than others. Lowland South America again illustrates some of the problems involved. A widespread, nearly defining feature of kinship systems in this region is a "Dravidian" division between (culturally defined) consanguines and affines coincident with a division between parallel and cross kin [57]. The problem for theories of kinship-as-reciprocity is that the two exchanging sides are not always meaningful social actors, and their "reciprocity" may be purely notional. The two sides may be categorized as moieties, but moieties *per se* may not be in the business of actually arranging marriages. Or the two sides may not receive native categorization at all. They may be more evident to the anthropologist collecting genealogies than to her informants [58].

Also, members of two sides, mutually consanguine and affine, commonly regard themselves not just as partners in reciprocity, but also as part of a community united by shared kinship. Regarding lowland South America [54]:

> Since siblingship commonly provides the dominant metaphor for solidarity and cooperation, relatives who are potential or actual affines may be viewed in different ways at different times – in some contexts, their distinctiveness from siblings is emphasized, but in others, the moral correlates of siblingship are generalized to the entire domain of kin relations, and the kin/affine opposition is transcended, or neutralized.

This quotation comes in the introduction to a collection of articles on the sibling relationship in lowland South America. The articles amply document the importance of sibling ties, going beyond descent theory and alliance theory. By way of illustration, consider one case, the Culina of western Amazonia [59]. The Culina are organized in named, preferentially endogamous, localized groups, *madiha*. In the context of marriage, a *madiha* is bifurcated into reciprocally intermarrying sets of consanguines and affines. But in other contexts, the Culina regard all the members of the *madiha* as close consanguineal relatives,

> underscored by statements such as … "we are all real siblings." … The *madiha* forms the boundary of the extension of kinship; members of other *madiha*, even those with whom one may use a kin term, are not considered to be kin in this sense.

The ethic of siblingship

> expresses a complex set of assumptions about proper behavior between individual *madiha* members. These include obligations to provide political support and mutual assistance, to share food and possessions, and to be "mild" and passive in interaction in contrast to the "wildness" … of jungle animals and hostile non-Culina.

These principles are normative, not simply the expression of individual sentiment.

[N]o division between types of kin, or kin and affines is accepted as a rationale for behavior which violates these norms. … those who repeatedly fail to act as proper "siblings" risk charges of witchcraft.

Turning back to lowland South America more generally, and to population genetics, Amazonian horticulturalists often combine parallel kin exogamy with community endogamy. Marriage practices like polygyny and endogamy raise coefficients of relatedness in these communities [60], which may be further reinforced when communities fission along kinship lines [61]. Those classified as non-consanguines where marriage is concerned are often biological kin, and recognized as such.

Amazonian horticulturalists differ in this respect from hunter-gatherers. The latter mostly have *extensive* kin ties, spreading relatedness broadly. The former more often have *intensive* kin ties, concentrating relatedness more narrowly, with relatively high coefficients of relatedness within groups. In one sample of 24 Amazonian horticulturalist groups, with adult populations ranging from 6 to 118, median 43, average individual coefficients of relatedness ($r_H$) within groups range from .016 to .216, median .079 [62].

Even if social interactions were entirely dyadic, elevated coefficients of relatedness resulting from inbreeding would be expected to favor higher levels of kin altruism. For example, Table 1 (Humans/ inbreeding) shows that when the parents of two siblings are first cousins – a frequent outcome of socially enforced marriage rules – the individual coefficient of relatedness ($r_H$) between the siblings is .56. But when inbreeding is combined with socially enforced nepotism, there is a potential for even more in the way of kin altruism. When three siblings whose parents are first cousins (Table 1/ Humans/ inbreeding, intensive kinship) play the Brothers Karamazov Game with equal helping abilities and no cheating, the effective coefficient of relatedness of two siblings toward a third ($r_G$) is .72.

And, to the extent that individuals – concerned with proper behavior and their standing in the community – abide by an ethic of siblingship, the potential for altruism is considerable. Table 1 (Humans/ inbreeding, intensive kinship) gives effective coefficients of relatedness ($r_G$) for Amazonian horticulturalists (see above) assuming that the adult community as a whole can enforce normative nepotism on its members. (Calculations follow [36]. To allow for differences in helping ability, the number of helpers, $n$, is set to half the adult number.) The median effective coefficient of relatedness among these groups is .54. Out of 24 groups, 15 have $r_G$ greater than .5.

The social anthropology of kinship is replete with examples of cultures defining social kinship in ways that depart systematically from kinship based on individual coefficients of relatedness. In some societies, social kinship is dialed up, or down, for purposes of regulating marriage, as different categories of relatives are defined as too related, or sufficiently unrelated, to marry [63]; this can affect individual relatedness, $r_H$, in the next generation. And in some of the same societies, in other contexts, both consanguines and affines are treated as closer kin, embraced by an ethic of siblingship. In these contexts, social kinship is dialed up, potentially fostering socially enforced nepotism in line with $r_G$, which can be greater than $r_H$. In this way, the cultural construction of kinship may serve to amplify genetic kin altruism.

## Supporting information

**S1 File.  Diploid brothers Karamazov game.**
(NB)

**S2 File.  Brothers Karamazov game, repeated prisoner's dilemma.**
(DOCX)

## Acknowledgments

The author thanks Juan Camilo Niño Vargas, Stephen Beckerman, and Manuel Lizarralde, organizers of the panel "Siblinghood in Lowland South America" at the 2023 14ᵗʰ Biennial Conference of the Society for the Anthropology of Lowland South America (SALSA), at which an early version of this article was presented. He thanks panel members for their comments and suggestions. He thanks Nadiah Kristensen [37] for her reanalysis of the original Brothers Karamazov Game in a novel framework, and for further discussion, which helped to prompt the analysis here. The editor and two reviewers suggested valuable revisions.

## Author contributions

**Conceptualization:** Doug Jones.

**Formal analysis:** Doug Jones.

**Investigation:** Doug Jones.

**Methodology:** Doug Jones.

**Software:** Doug Jones.

**Writing – original draft:** Doug Jones.

**Writing – review & editing:** Doug Jones.

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
