## [Decision Letter · Decision Letter 0]

8 Dec 2024

PONE-D-24-41944Extraordinary Siblings: Mole Rats, Marmosets, and Radcliffe-BrownPLOS ONE

Dear Dr. Jones,

Thank you for submitting your manuscript to PLOS ONE. After careful consideration, we feel that it has merit but does not fully meet PLOS ONE’s publication criteria as it currently stands. Therefore, we invite you to submit a revised version of the manuscript that addresses the points raised during the review process.

Both reviewers felt that the paper had value, as do I. However, both reviewers raised some concerns that need to be addressed before this manuscript could be accepted. I will not reiterate all their comments here, but I do want to highlight the comment(s) about the presentation of the mathematical models and how they should be presented in a more accessible way. This is important not just to help reviewers assess the model, but crucially to help readers who aren’t math modelers follow your logic. As Reviewer 1 notes, readers will understand your argument better if you can break down the math a bit more for them. And the better that readers can understand your argument, the more they’ll cite your arguments. As such, I request that you do more to walk readers through the model, and how the equations were derived, with or without numerical examples, to make the logic clearer.

Both reviewers also noted that you can do more in the introduction to set up your main takeaway (Reviewer 1) and to distinguish this work from your previous work (Reviewer 2). PLOS ONE publishes replications and does not rely on novelty, but more readers will be excited about this paper if you can show (and show early) why this paper is novel and worth reading beyond what you’ve already published.

In addition to the reviewers’ comments, I have two additional comments. First, I notice at least one reference to Jonathan Pruitt’s work. He has multiple allegations of data fabrication against him, including multiple papers being retracted after investigation and at the request of his co-authors. He has since left academia as a result of these allegations. At this point, most or all of his papers are under a cloud of suspicion, because we don’t know which of them – if any – are based on legitimate analyses of real data. This greatly weakens any argument that relies on a citation of Pruitt. Please reconsider these citations – find a different paper to make the same point.

Second, in Table 1, it’s easy for readers to misunderstand your argument about effective r’s in humans. Given that this is the introduction, readers could very easily misinterpret your argument to say that (for example) sibling r=.67 with 3 siblings under normal circumstances. Instead, these effective r’s rely on the arguments that you make later in the paper – r would not be that high without it. Is there a way you can do more to drive home that this extraordinary relatedness is not a starting point, but instead is the outcome of your model? For example, can you add a quick note to where in your paper these are derived? Example: “Humans *Homo sapiens*
(in my model) ” and “reputation + norms → socially enforced nepotism (Section X.X) ”. Incidentally, this is one minor way in which you can sell the importance of this paper early, related to the reviewers' point about setting up your main takeaway earlier.

We look forward to receiving your revised manuscript.

Kind regards,

Pat Barclay

Academic Editor

PLOS ONE

Journal Requirements:

Reviewers' comments:

Reviewer's Responses to Questions

**Comments to the Author**

1. Is the manuscript technically sound, and do the data support the conclusions?

Reviewer #1: Yes

Reviewer #2: Yes

2. Has the statistical analysis been performed appropriately and rigorously? 

Reviewer #1: N/A

Reviewer #2: N/A

3. Have the authors made all data underlying the findings in their manuscript fully available?

Reviewer #1: Yes

Reviewer #2: Yes

4. Is the manuscript presented in an intelligible fashion and written in standard English?

Reviewer #1: Yes

Reviewer #2: Yes

5. Review Comments to the Author

Reviewer #1: The manuscript uses the Brothers Karamazov Game to explore the idea that, through the use of social norms, humans are able to create "extraordinary siblings," i.e., siblings with "effective genetic relatedness" of greater than 0.50.

The intellectual quality of the manuscript is high, and the topic is interesting. For those reasons, I support its eventual publication. However, without a fairly major rewrite few readers will be able to understand it, and it will then not receive the attention it deserves.

The various formulas and associated mathematics are particularly hard to follow. I recommend that the author help readers out by plugging some numbers into those various equations and showing how they work. The numbers could come from stories about altruism toward kin. Even if the stories are cartoonish and unrealistic, they would help the reader understand the author's point.

The section on South America is very vague and general. Can the author please provide some more details, including the names of the ethnic groups mentioned? Interested readers should not have to dig up the references the author cites in order to figure out what those groups are and how their social systems work.

Early in the manuscript, explain more fully what "effective r" is and how it differs from regular r.

The idea that altruism toward kin is a public good is crucial, but it doesn't appear until the results section. I suggest explaining it somewhere in the introduction, instead. That explanation should not assume that readers know what a public good is and what the implications are when something is a public good. Explain the characteristics of public goods (low subtractability, difficult exclusion) and explain how those characteristics can lead to a collective action dilemma, also known as a social dilemma or free-rider problem.

Reviewer #2: The paper presents the Brothers Karamazov Game as a means for exploring solidarity between siblings. It was difficult from this paper to determine how this advances previous papers about the Karamazov game. Also, the game nicely opens up possible strategy spaces beyond existing games. However, to be convincing, I think it would require a careful analysis of evolutionary stability with a range of possible strategies.

Jones, D. (2000). Group nepotism and human kinship. Current Anthropology, 41(5), 779-809.

Jones, D. (2004). The universal psychology of kinship: Evidence from language. Trends in Cognitive Sciences, 8(5), 211-215.

Jones, D. (2016). Socially enforced nepotism: How norms and reputation can amplify kin altruism. PLoS One, 11(6), e0155596.

6. PLOS authors have the option to publish the peer review history of their article (what does this mean? ). If published, this will include your full peer review and any attached files.

**Do you want your identity to be public for this peer review?** For information about this choice, including consent withdrawal, please see our Privacy Policy .

Reviewer #1: No

Reviewer #2: No

---

## [Decision Letter · Decision Letter 1]

3 Feb 2025

Extraordinary Siblings: Mole Rats, Marmosets, and Radcliffe-Brown

PONE-D-24-41944R1

Dear Dr. Jones,

We’re pleased to inform you that your manuscript has been judged scientifically suitable for publication and will be formally accepted for publication once it meets all outstanding technical requirements.

There is one remaining change that will improve the manuscript and make it easier for readers to understand. In Table 1, put baseline relatedness for humans without any factors as a comparison group, and label it as such. Then for anything that relies on your paper, put in brackets “(this paper)” in the final column to drive home that this is derived from your result, not from the background literature. This helps to highlight the contributions of this manuscript: it shows that the effective relatedness you present is not a "given" or background knowledge - it's something that you will derive later in the manuscript, and readers should keep reading to see how you derived it. This change is very minor and can be done in the subsequent stages, which is why I am accepting the manuscript now.

Kind regards,

Pat Barclay

Academic Editor

PLOS ONE

Additional Editor Comments (optional):

Reviewers' comments:

Reviewer's Responses to Questions

**Comments to the Author**

1. If the authors have adequately addressed your comments raised in a previous round of review and you feel that this manuscript is now acceptable for publication, you may indicate that here to bypass the “Comments to the Author” section, enter your conflict of interest statement in the “Confidential to Editor” section, and submit your "Accept" recommendation.

Reviewer #1: All comments have been addressed

2. Is the manuscript technically sound, and do the data support the conclusions?

Reviewer #1: Yes

3. Has the statistical analysis been performed appropriately and rigorously? 

Reviewer #1: Yes

4. Have the authors made all data underlying the findings in their manuscript fully available?

Reviewer #1: Yes

5. Is the manuscript presented in an intelligible fashion and written in standard English?

Reviewer #1: Yes

6. Review Comments to the Author

Reviewer #1: The manuscript is much easier to understand now. I appreciate the work the author put into it in order to improve it.

7. PLOS authors have the option to publish the peer review history of their article (what does this mean? ). If published, this will include your full peer review and any attached files.

**Do you want your identity to be public for this peer review?** For information about this choice, including consent withdrawal, please see our Privacy Policy .

Reviewer #1: No

---

## [Editor Report · Acceptance letter]

PONE-D-24-41944R1

PLOS ONE

Dear Dr. Jones,

I'm pleased to inform you that your manuscript has been deemed suitable for publication in PLOS ONE. Congratulations! Your manuscript is now being handed over to our production team.

Kind regards,

on behalf of

Dr. Pat Barclay

Academic Editor

PLOS ONE